# Thiyl Radicals: Versatile Reactive Intermediates for Cyclization of Unsaturated Substrates

**DOI:** 10.3390/molecules25133094

**Published:** 2020-07-07

**Authors:** Dylan M. Lynch, Eoin M. Scanlan

**Affiliations:** School of Chemistry and Trinity Biomedical Sciences Institute (TBSI), Trinity College Dublin, The University of Dublin, Dublin 2, Ireland; dylynch@tcd.ie

**Keywords:** radical, cyclisation, thiyl radicals, sulfur, carbocyclization

## Abstract

Sulfur centered radicals are widely employed in chemical synthesis, in particular for alkene and alkyne hydrothiolation towards thioether bioconjugates. The steadfast radical chain process that enables efficient hydrothiolation has been explored in the context of cascade reactions to furnish complex molecular architectures. The use of thiyl radicals offers a much cheaper and less toxic alternative to the archetypal organotin-based radical methods. This review outlines the development of thiyl radicals as reactive intermediates for initiating carbocyclization cascades. Key developments in cascade cyclization methodology are presented and applications for natural product synthesis are discussed. The review provides a chronological account of the field, beginning in the early seventies up to very recent examples; a span of almost 50 years.

## 1. Introduction

Since the turn of the millennium, organosulfur compounds have garnered substantial interest in the fields of medicinal chemistry, chemical biology and material science [1,2,3,4,5,6,7]. Of the diverse biochemical processes involving free radicals, many are thiyl radical mediated, highlighting the chemoselectivity, efficiency and biocompatibility of these reactive intermediates [8]. Sulfur-centered radicals have been broadly utilized in a wide range of synthetic applications over the past three decades. While their general utility has been previously reviewed [9,10,11,12,13,14,15,16,17], this account seeks to update the reader specifically on the application of thiyl radicals (RS^•^), in the carbocyclizations of unsaturated moieties, and the scope and limitations of these reactions in natural product synthesis.

Sulfur, in its −2 oxidation state, can act as either a nucleophile or as a radical. The S-H bond is generally considered relatively weak, with a bond dissociation energy (BDE) of 365 kJ/mol [18], and as such, both alkyl thiols and thiocarboxylic acids (thioacids) are precursors to thiyl/acyl thiyl radicals through facile homolytic cleavage. Thiyl radicals may be generated by homolytic bond breakage, and by one-electron redox processes. In the former process, a carbon centered radical (for example, from the radical initiator azobisisobutyronitrile (AIBN)) will abstract a hydrogen atom from the thiol. Rate constants for this homolytic breakdown are of the magnitude 10^8^ M^−1^ s^−1^, which is below the diffusion limit for this reaction [19]. In the case of redox processes, single electron oxidants such as Mn (III) complexes or other metal sources can be used to accomplish radical formation [20,21,22]. Both radiolysis and direct photolysis of the S-H bond under UV irradiation are also valuable pathways to thiyl radical formation [23,24].

Thiyl radicals are well known to react efficiently at sites of unsaturation, through the thiol-ene and thiol-yne reactions. A full analysis of the utility of these reactions is beyond the scope of this review, and readers are directed to the reviews already present in the literature [6,24,25,26,27,28,29]. This account will focus exclusively on the intramolecular trapping of the nascent carbon radical formed via thiyl radical addition, and how these radicals form rings and products in cascade reactions.

## 2. Dienes and Diynes

### 2.1. Origin 

For the purpose of this review, ‘homogenous’ infers the same degree of unsaturation for all reactive sites in the substrate or substrates; i.e., dienes, diynes, trienes, etc. Possibly the earliest example of a cyclization cascade on a homogenous substrate was published by Kuehne et al. in 1977 [30]. The account details the cyclization of selected dienes with a range of thiyl radical precursors, relying upon radical initiation via photolysis of diphenyl disulfide or thermolysis of benzoyl peroxide. Kuehne and co-workers sought to compare the cyclizations effected by a range of representative thiyl radical species, and thus experimented with the initiator itself, and thermal versus photoinitiation. While they obtained a number of acyclic adducts and complex mixtures of stereoisomers, a number of cyclized products were also identified. The representative example in Scheme 1 shows the cyclization of dimethyl diallylmalonate **1** under a number of reaction conditions, the best of which employed the photolysis of diphenylsulfide in benzene to generate the thiyl radical of ethanethiol. These conditions resulted in the formation of over 90% cyclized product **2**, and six minor acyclic adducts. However, the formation of any six-membered cyclization products was ruled out systematically.

Kuehne and co-workers applied these findings to the analogous cyclization of α-acoradiene **4**, which proceeded almost quantitatively. Desulfurization with Raney nickel yielded dihydrocedrene **6**, the hydrogenated form of a sesquiterpene found in the essential oil of cedar (Scheme 2). This approach was later applied to the cyclization of geranyl acetate, but the acyclic monoadducts of geranyl acetate and the thiyl radical source were mostly obtained.

These results were greatly expanded upon in work published by Padwa et al. in 1985 [31]. Using thioacetic acid and AIBN, Padwa and co-workers were able to cyclize a relatively simple tosylamine **7** with two allyl substituents, in a 5-*exo-trig* process. Addition of the thiyl radical, formed indirectly by thermolysis of AIBN, and intramolecular cyclization of the nascent carbon-centered radical **8** immediately proved the utility of this reaction as a viable route to heterocycles (Scheme 3A). It may be expected that the *trans* isomer would dominate in this process, whether for steric reasons or otherwise, but in fact a general preference for the *cis* isomer is observed. Hyperconjugative mixing of the half-filled p orbital and the C-H σ and σ* orbitals produce a semioccupied delocalized orbital which is of matching symmetry to the alkene π* orbital. Therefore in the transition state, the *cis*-isomer exhibits a second interaction between the carbon-centered radical and remaining olefin, offsetting the nonbonded repulsion between the vicinal substituents of the newly formed pyrrolidine [32,33,34]. Simple substitution at the alkene terminus does not affect the cyclization, and the process continues to follow a 5-*exo*-*trig* pathway.

Padwa et al. also reported the addition of an acetylthiyl radical onto another homogenously unsaturated substrate; the corresponding *N*,*N*-dipropargylsulfonamide **11** (Scheme 3B). In this case however, a crystalline solid was obtained from the reaction as the major product, which was later identified as the thienopyrrole **14**. Following addition of the thiyl radical to the alkyne, the nascent carbon-centered radical cyclizes as shown in Scheme 3. However, this results in formation of a highly unstable vinyl radical intermediate **13**, which through an intramolecular homolytic substitution (S_H_i) extrudes an acyl radical to afford the observed product **14**. Indeed, this is one of the few examples of a diyne cyclization via thiyl radicals that one can find in the literature. A second account of diyne reactivity under thiyl radical addition appeared three decades later, in which Agrawal et al. optimized a number of substrates with malonate-type backbones for cyclization to the respective thiophene [35]. This approach relies on thermolysis of AIBN in toluene with thioacetic acid as the acyl thiyl radical source, and furnished a number of the bicyclic thiophenes, in good yield.

### 2.2. Early Applications in Total Synthesis

Shortly thereafter, a significant advancement in the application of thiyl radical initiated cascades was published by Naito et al. [36]. In a publication focused on the total synthesis of two imidazole alkaloids, Naito and co-workers employed the thiyl radical addition/cyclization cascade of an *N*-allyl-*N*-benzylcinnamamide **15** with diphenyl disulfide and thiophenol in equimolar quantities, which under photochemical irradiation cyclized to the desired 3,4-disubstituted pyrrolidinone **16**, albeit as a racemic mixture. This afforded the necessary building block to complete their synthesis of isoanatine **17** and anantine **18** (Scheme 4A) [37]. Just two years later, Naito et al. expanded upon the utility of this reaction, in part one of a series on radical cyclization in heterocycle synthesis [38]. Through application of the methodology developed for their previous alkaloid synthesis, Naito and co-workers developed a new synthetic method which furnished nitrogen-containing heterocycles from simple dienylamide starting materials. This addition generates either one or two carbon-sulfur bonds, and the desired carbon-carbon bond. The authors sought to establish the regiochemistry of thiyl radical addition, the cyclization mode (i.e., 5-*exo*-*trig* vs. 6-*endo*-*trig*) and finally, whether any stereochemical control was exerted by the process. By using a substrate with two different types of olefins, an insight into the chemoselectivity was also recorded. 

It was postulated that the cyclization occurred through addition to the acrylamide terminus, followed by cyclization onto the alkene of the neighboring allyl substituent, through a 5-*exo*-*trig* pathway (Scheme 4B). The phenylthiyl radical generated in this reaction displays reactivity somewhere between a nucleophilic and electrophilic radical [39], therefore the SOMO should react more favorably with the lower energy LUMO of an electron deficient alkene—in this case the acrylamide substituent. Furthermore, as the thiol-ene reaction is reversible [40,41] the concomitant radical formed via addition to the acrylamide terminus **20** would be further stabilized by its adjacent carbonyl group. In the case of thiyl radical addition to the allylic terminus **23**, this stabilization is absent and as such the equilibrium of the thiol-ene reaction favors the starting amide. Naito and co-workers went on to examine the substituent effects, and found that those substrates with no additional functionality at the β-position of the acrylamide group primarily furnished lactams. The presence of a substituent α to the unsaturated amide appeared to have no effect on cyclization dynamics. However, in the case where both olefin termini are substituted, even with a methyl group, no cyclization was observed.

Shortly before the turn of the millennium, another example highlighting the efficacy of the thiyl radical addition/cyclization process emerged from Harrowven and collaborators [42]. In the course of their work in the terpenoid field, the authors took advantage of a sulfur mediated radical cyclization strategy to complete the total syntheses of aplysin and debromoaplysin. Through clever retrosynthetic analysis, Harrowven et al. identified that a 5-*exo*-*trig* cyclization through a chair-like transition state would simultaneously furnish both the sterically demanding tricyclic backbone of aplysin, and generate the relative stereochemical configuration for three neighboring stereogenic centers. Employing a Raney nickel desulfurization strategy, Harrowven et al. obtained the necessary skeleton **29** in a diastereomeric ratio of 8:1 (Scheme 5). Similar to the previously discussed strategy employed by Kuehne et al., the use of thiyl radicals offers a much cheaper and less toxic alternative to the archetypal organotin based radical methods.

From this point, Harrowven and co-workers delved deeper into the dynamics of the thiyl mediated cyclization and exploited the reaction for the cyclization of 1,6-dienes. A further account published by Harrowven et al. [43] supersedes the earlier aplysin work, and a diverse library of thiabicyclo[3.3.0]octanes were synthesised through the co-cyclization of 1,6-dienes with concomitant sulfur atom transfer. The authors proposed that if the rules provided by Beckwith for ring closure [44,45] were obeyed, a second cyclization (following initial diene cyclization), of a radical intermediate predisposed to homolytic substitution at sulfur would be observed. In their screening of reaction conditions, Harrowven et al. identified that photolysis of a solution of the diene and di-*tert*-butyl disulfide in hexanes provided the desired tetrahydrothiophene; reactions only reached an appreciable efficiency when a quartz photochemical cell was employed (instead of a Pyrex cell), and addition of triethylborane greatly accelerated the rate. Up to 31% starting material was recovered in some cases, yet the tolerance of the thiyl radical addition/cyclization is evident in the examples shown below (Scheme 6). Insert A shows some of the more exotic examples furnished by Harrowven and co-workers. In a follow up letter Harrowven et al. expanded the cyclization to a solid support [46] using a carbodiimide coupling to a Wang-type resin [47], noteworthy in that very few tin-free radical cyclizations have been conducted on solid supported substrates [48,49,50]. Relying instead on thermolysis of AIBN for initiation, the authors demonstrated conservation of both yield and diastereoselectivity with respect to their preceding solution phase results. Further utility of the reaction in terpenoid synthesis by Barrero et al. [51] involved a short synthesis of (±)-dehydroiridomyrmecin, a natural iridoid. Again using two equivalents of thiophenol and one equivalent of AIBN in refluxing benzene afforded the backbone for this biologically interesting iridane, and comprised the first total synthesis of this substrate.

### 2.3. Dienes as Precursors to Heterocycles.

Miyata et al. published two brief accounts on the application of thiyl radical addition/cyclization; these accounts comprise parts ten and eleven in a series on heterocycle synthesis via radical cyclization from the Naito group [52]. The first of these accounts employs a familiar cyclization strategy, but includes a clever elimination step (Scheme 7) which permits the use of a catalytic amount of the thiol (thiophenol). Miyata and co-workers designed a sequential formation of two bond and one bond cleavage, which furnished a trisubstitutued pyrrolidine ring **39**. This methodology was implemented by the authors in the total synthesis of the marine product (−)-α-kainic acid **40**, a potent neuroexcitatory amino acid agonist with central nervous system activity [53]. Good yields were obtained using low loading of equimolar thiophenol and AIBN.

A later example by Miyata et al. [52] features a variation on previous approaches, many of which involve backbones composed mainly of carbon (Scheme 8). Using dienes connected via hydroximates **41**, the authors reported a convenient conversion of cyclic hydroximates formed from thiyl radical addition/cyclization, into the corresponding α,β-disubstituted-γ-lactones. The hydrolysis of the cyclic hydroximates gave the desired *cis*- and *trans*-lactones in excellent yield, and the methodology was employed in the synthesis of (±)-oxo-parabenzlactone. Interestingly, the unstable *cis*-isomer **43** formed via cyclization could be irreversibly resolved into the synthetically more valuable *trans*-isomer **42** by treatment with sodium ethoxide.

Miyata et al. [54] also published a wealth of cyclizable diyne systems, in their “sulfanyl radical addition-addition-cyclization” (SRAAC) strategy. The unbranched diynes discussed herein yield *exo*-olefins upon completion of the cascade, and those diynes possessing quaternary carbons furnished cyclized *endo*-olefins (Scheme 9, *vide infra*). The authors applied this chemistry to the synthesis of a small fragment of 1α,25-dihydroxy vitamin D_3_. In agreement with results published by their contemporaries, the 1,8-diyne species failed to cyclize under a variety of conditions.

Pedrosa et al. [52] shortly thereafter reported on the carbocyclizations of perhydro-1,3-benzoxazines, furnishing 3,4-disubtitued pyrrolidinone derivatives with good diastereoselectivity and complete regioselectivity. A wide array of substituted products were synthesized, and substitution at both olefin termini was well tolerated. The author’s results further supported the findings of Miyata et al. [55], in that a phenyl thioether **62** suitably disposed to radical elimination can generate secondary olefins **64** as a result of the carbocyclization process (Scheme 10). The preference of the thiyl radical SOMO to interact with electron deficient alkenes was also exploited, with the nascent phenythiyl radical attacking the acrylic olefin before cyclizing through the expected 5-*exo*-*trig* pathway onto the substituted alkene. 

### 2.4. Moving Forward

At this juncture, the literature is composed of further accounts on the tolerance of the thiyl radical addition/cyclization process to varying functional groups (FGs), and steric demands. Hodgson et al. [56] demonstrated the utility of this process to access sterically challenging substrates, and synthesized a number of 7-thio-substituted norbornenes from norbornadienes (Scheme 11A). The diene cyclization product **67** appeared to dominate over radical quenching from solvent or otherwise, with a ratio of 1:6 of **66**:**67** obtained. This work also constitutes one of the first examples in this field of *3-exo-trig* cyclization, further highlighting the potential of thiyl radicals to access sterically demanding substrates. A communication by James et al. [57] investigated the tolerance of the cascade in question to 3-silylheptanyldiene systems (Scheme 11B). While sulfur-mediated cyclizations only composed a short portion of this work, a number of substituted dienes were successfully cyclized, providing the familiar thiabicyclo[3.3.0] skeleton in one step due to the S_H_i mechanism previously discussed.

Kamimura and co-workers published an account [58] on the application of radical cascades for the synthesis of the oxa-tricyclic core of platensimycin **72**, an antibiotic that inhibits two enzymes in fatty acid biosynthesis (Scheme 12). In a concise account from 2015, the authors describe a short thiyl radical triggered cyclization in their attempts to form the multicyclic core, and while their cyclization of diallymethylene-δ-lactone **73** proceeded in a highly stereoselective manner, the incorrect stereochemistry was obtained.

Wang and colleagues have also been active in this area, and published a tandem sulfenylation/cyclization of *N*-arylacrylamides through a 5-*exo*-*trig* mechanism, which boasted excellent yields and control of regioselectivity for certain substrates (Scheme 13) [59]. This methodology furnished a number of 3-(sulfenymethyl)oxoindoles **77** and 3-sulfenyl-3,4-dihydroquinolin-2(1H)-ones **78**. Up to this point in the literature, the vast majority of thiyl radical precursors have been the respective thiols, thioacids or disulfides; this account offers an unconventional precursor to aryl thiyl radicals. The authors propose a mechanism in which iodine functions as an oxidant, reductant, and radical initiator, to generate aryl thiyl radicals from the corresponding sulfonyl hydrazides **76**. This offers an accessible route to thiyl radical addition/cyclization as sulfonyl hydrazides are generally moisture-compatible solids, lack the expected odour and are amenable to handling by organic chemists outside traditional ‘sulfur labs’. This methodology permits the cyclization of α,β-disubstituted acrylamides, substrates which have previously been found recalcitrant to cyclization. It is noteworthy that the α,β-unsubstituted acrylamides failed to cyclize, instead furnishing bisthioethers.

An example which highlights the significant potential of the thiyl radical addition/cyclization in diene cascade reactions comes from Hashimoto et al. [60]. This publication from the Maruoka group details an organic thiyl radical catalyst for enantioselective cyclization (Scheme 14). Through the careful design of a chiral pocket around the chiral thiol precatalyst, the authors were able to exact an enantio- and diastereoselective C-C bond forming cyclization with concomitant extrusion of the thiyl radical at the end of the cascade. With 3 mol% loading of a binaphthyl-modified catalyst and benzoyl peroxide to generate the thiyl radical, a wide array of substrates were successfully cyclized by this catalyst, in yields up to 99% and diastereomeric rations of up to 95:5. Indeed the catalyst loading could be scaled as low as 1 mol% at larger scales (1 mmol) [61]. In their studies to optimize catalyst structure, Hashimoto et al. were able to attain yields of up to 95%, with a diastereomeric ratio (d.r.) of 95:5 and enantiomeric excess (e.e.) of 86% using sunlight as the photolysis source. Hashimoto et al. continued to develop this methodology, and in 2016 published a communication on the use of bulky thiyl radical catalysts for [3 + 2] cyclizations, this time with *N*-tosylvinylaziridines and alkenes [62].

In a further example of the application of thiyl radicals within organocatalytic systems, work analogous to that of Hashimoto et al. was published by Ryss et al. (Scheme 15) [63]. This study detailed the use of disulfide-bridged peptides **95** to mediate enantioselective ring-opening and cycloaddition of vinylcyclopropanes. Using photolysis to cleave the disulfide bridge of a suitable cysteine-based dimeric peptide, two equivalents of the thiyl radical per equivalent of catalyst were generated *in situ*. The authors provided an extensive analysis of potential backbone interactions between their malonic acid scaffold and the catalyst, and found that while the malonic esters provided mostly racemic products, the corresponding amide-substituted vinylcyclopropanes **96** were more successful in the discussed transformation. This was presumed by Ryss et al. to originate from hydrogen bonding interaction between the peptide catalyst and the substrate. In their design of the catalyst the appendage attached to the proline backbone was altered; one example (in blue) is shown in Insert A below.

The use of thiyl radical addition/cyclization has been previously described as a useful route to thiophenes, through clever use of homolytic substitution reactions. Possibly the most recent cyclization of a diyne substrate was reported by Dutta et al., in which a number of pyrroles were prepared (Scheme 16) [64]. This account describes the use of ynamides and alkynes for cyclization; the authors report a ‘serendipitous observation’ that the reactivity of simple alkynes was found to exceed the traditionally more reactive ynamides moieties for cyclization. This led to the development of a methodology in which an alkyne **99** is first attacked by the aryl sulfenyl radical formed from **100**, which then cyclizes in a 5-*exo-dig* pathway onto the ynamides to generate a number of novel 4-thioaryl pyrroles **101**, **103**–**110**. The methodology tolerates an array of functional groups and *N*-protecting groups, and forbearance to modification of the ynamides and propargylic terminus was also recorded.

Thus far, this review has shown numerous examples that have employed the thiyl radical addition/cyclization cascade for a variety of substrates. But what are the limitations of the process? A 2017 publication [65] by Wang and colleagues poses an interesting question—is it possible that radical polycyclization could yield dodecahedrane **112**, one of the top synthetic targets sought by organic chemists for decades [66]? Woodward failed in synthesizing this Platonic hydrocarbon [66,67], but the Paquette group succeeded the first synthesis in the early 1980s [68]. The account by Wang et al. assesses the viability of thiyl radicals as mediators of polycyclizations to construct complex molecular architectures (Scheme 17). Using density functional theory (DFT) calculations, the authors concluded that the process is energetically viable in both a kinetic and thermodynamic sense. While the reversibility of the thiol-ene and potential quenching by hydrogen atom transfer were highlighted, the quantum chemical calculations indicate the possibility that one day, these complex geometries may be obtained using simple thiyl radical cascades.

## 3. Heterogeneous Substrates: Enynes and Other Mixed Systems

### 3.1. Origin

The two earliest examples of thiyl radical cascades orchestrated on heterogeneous or ‘mixed’ systems were both reported in 1987 (Scheme 18). The first of these sulfur-triggered cyclizations, reported by Broka et al. [69], in which nascent vinyl radicals **114** formed via thiyl addition to enynes **113**, regioselectively cyclized to furnish the respective cyclohexylidine **115**. The authors identified that under ionic conditions (in this case, refluxing thiophenol with the enyne in benzene), very little cyclization takes place as the rate of hydrogen abstraction by the vinyl radical outcompetes the cascade reaction. By switching to a procedure with AIBN, and slow addition of the thiol, the authors were able to favor cyclization over quenching of the radical cascade. This gave **115** as the cyclized product, in a 75% yield when 2,2,5,5-tetramethyltetrahydrofuran (TMTHF) was used as the solvent. Broka et al. proposed that equilibration of the cyclopentylmethyl radical, formed via 5-*exo*-trig cyclization of **114**, to the more stable cyclohexyl isomer prior to hydrogen abstraction facilitates the generation of **115.** A number of other systems were tested to examine the generality of enyne cyclization, and it was found that as the substitution of the olefin increased, the success of cyclization severely diminished; for example, enyne **119** gave only 33% cyclized products, and substrate **121** gave none.

Ichinose et al. reported an account of triethylborane based radical initiation for addition to acetylenic compounds. The authors explored addition of thiyl groups to terminal acetylene compounds [70], which proceeded regioselectively but not stereoselectively (Scheme 19). The three examples of interest to this review are that of an intermolecular dihydrothiophene synthesis **129**, and the vinyl radical cyclization of two acetylenic olefins **130** and **132**. Analogous to the work carried out by Broka et al. in the same year, a clear trend in the substitution and steric demands around the olefin was identified.

### 3.2. Broadening the Scope

Broadening the scope of the review to consider ‘mixed’ unsaturated systems, the inclusion of unsaturation at heteroatomic moieties must also be considered. A very early example form El Kaim et al. [71] offers 5- and 6-*exo*-*trig* cyclizations of oxime ethers and hydrazones. By generating thiyl radicals from thiophenol, a number of ethylenic and acetylenic hydrazones were cyclized under thermal initiation with AIBN in refluxing cyclohexane (Scheme 20). Analogous to early work by Padwa et al. previously discussed, the *cis* isomer predominates in the cyclization, despite the inclusion of heteroatoms. Furthermore, no trace of uncyclized product was detected for the conditions reported herein. The methodology published in this account widened the scope for cyclizations on the C=N bond, and general examples from this work are detailed below to show the range of functional group tolerance. Despite this clear forbearance to varying functionality, example **138** did not cyclize.

In an account by Alcaide et al., an early medicinal chemistry application of the thiyl radical addition/cyclization cascade is described [72]. Highly functionalized rings fused to a common pharmacophore, a β-lactam, were synthesized via radical cascade chemistry on a number of Baylis-Hillman adducts, providing the basis for this stereocontrolled and chemoselective approach shown in Scheme 21. Several phenylthiovinyl derivatives **143a**–**c** of the yne-ene based β-lactams **142** were obtained, and subject to further transformation. It should be noted that in the case of *n* = 4 i.e., the 5-hexynyl conjugated β-lactam, neither the thiyl radical addition nor classical methods such as organotin chemistry could cyclize the substrate. It is also noteworthy that it was possible for the authors to initiate thiyl-yne conjugation prior to the Michael addition, which one would expect to compete for the thiol via thia-Michael addition.

With each publication that bolsters the robustness of thiyl radical utility in organic synthesis, further technologies such as flow chemistry and microwave assisted synthesis become available to the sulfur chemist. In one such paper on the use of microwave assisted radical cyclizations, Lamberto et al. [73] discuss the use of a wide variety of thiols to generate pyrrolines and pyroglutamates from alkenyl and alkynyl isocyanides (Scheme 22). This use of microwave ‘flash-heating’ technology boasts the advantage of significantly faster reaction times, and increased yields. It is worth noting that in the case of alkynyl isocyanides, protection of the alkyne terminus as a silyl ether was necessary, as unprotected substrates suffered from severely diminished cyclization yield. In a further publication [73] the authors demonstrated the applicability of this methodology to solid phase synthesis, and furnished pyroglutamates and 2-mercaptopyrrolines from three polymer-supported isocyanides using commercially available resins.

In the mid-2000s, a relative ‘renaissance’ in the use of thiyl radicals as instigators of carbocyclizations was observed. While the present review is far from exhaustive, a description of the main outcomes from key research groups will be discussed. Majumdar and co-workers have been particularly active in this area, publishing heterogeneous cyclizations for a range of heterocycles, including cyclic ethers **157**, pyrimidine-fused azocine derivatives [74], furo- and pyranocoumarin species [75], a range of pyrrolopyrimidines as 9-deazaxanthine analogs **163** [76], as well as benzoxocine **151 [77]** and oxepin derivatives [78]. In several publications on the topic of heterogeneous carbocyclizations between 2007 and 2010, Majumdar et al. have reported a number of difficult annulations, and broadened the role of thiyl radicals to catalyzing Claisen rearrangement, and undergoing 8-*endo* cyclizations for accessing sesquiterpenes. In the interest of space constraints, these accounts will not be described in full detail herein and instead a selection of key achievements is shown in Scheme 23.

Montevecchi et al. detailed the addition of thiyl radicals to alkynyl azides [79] as well as attempted cyclizations onto carbonyls and cyano groups [80]. The group continued to explore these processes, and also published on the relative reaction rates of thiyl radical addition and hydrogen atom abstraction [81]. In their investigation of alkynyl azides as suitable substrates for thiyl radical mediated cyclization, the authors identified two suitable 2-sulfanylvinyl radicals which underwent 5-*exo* cyclization onto the azido moiety, preventing intramolecular addition to the sulfanyl aromatic ring (Scheme 24**)**. In the course of this project, Montevecchi and co-workers rationalized the ability of these two radicals to cyclize by considering the resonance stabilization of the nascent triazenyl radicals, caused by delocalization into the indole ring. They also reasoned that the failure of other radicals to undergo cyclization may be explained by unfavorable polar effects in the transitions state between the more electrophilic vinyl radicals and the aliphatic azido moiety. Note that while benzyl mercaptan is shown as the thiol agent in Scheme 24, similar success was observed with thiophenol as the thiyl radical precursor.

Indeed, even if cyclization attempts fail and do not yield any desired product, these results are nonetheless valuable. In a follow up publication to their success with alkynyl azides, Montevecchi et al. identified several sulfanylvinyl radicals which did not undergo any 5-membered cyclization to esteric or thioesteric carbonyl moieties, nor did they obtain any evidence of successful toluenesulfanyl radicals addition to a number of alkynyl nitriles. Montevecchi and co-workers did, however, identify two radical intermediates which were successful in cyclizing onto aromatic cyano groups, which furnished a useful synthetic route to indenones.

### 3.3. Cyclizations onto Nitrogenous Moieties, and Further S_H_i Chemistry

An excellent example of the convenience of thiyl radical addition/cyclization cascades with heterogeneous substrates comes from Keck et al. who employed 6-*exo* cyclizations of substituted thiovinyl radicals with oxime ethers for the synthesis of high value alkaloids. In their total synthesis of (+)-lycoricidine **176** and (+)-narciclasine **179**, a nascent vinyl radical is generated via addition of a phenylthiyl radical to a disubstituted alkyne. Thiyl addition to the alkyne, and subsequent addition to an oxime ether furnishes the corresponding ring system **175** or **178**, which was then further transformed to the target alkaloid. In the course of these studies, the authors noted that no formal synthesis of narciclasine had been accomplished, and thus applied the same methodology with some modification to the substrate backbone (Scheme 25). Keck and co-workers also identified the reverse regiochemistry of stannyl and phenylthiyl radicals to the same alkyne in this account.

Fernández et al. published a one-step cyclization of ketimines into cyclized allylamine derivatives in the mid-2000s, which again relies upon a tandem radical addition/cyclization strategy (Scheme 26) [82]. Using a substrate inspired by the research group’s interest in tetrodotoxin, a number of thiyl radical mediated cyclizations were attempted. The authors sought to exert control over cyclization through the 6-*endo* versus the *5-exo* mode. Indeed by substitution the alkyne terminus with a phenyl group **185** restored some control over the cyclization mode, and through using the traditional PhSH/AIBN couple, a number of conditions were explored. Interestingly, the use of a medium pressure mercury lamp as the irradiation source permitted the use of just 1.05 equivalents of thiophenol and no AIBN was required; these conditions gave the highest cyclization yield of 75%. However this success was reliant on the alkynyl substituent being a phenyl group, as esters and ethers did not promote formation of the desired cyclohexeynlamine **186**.

Friestad et al. published an account of chiral hydrazone cyclization, using silicon tethered alkynes (Scheme 27) [83]. A one pot process involving thiyl addition to the ethynyl terminus and a desilylation with standard fluoride sources furnished (*E*)-vinylsulfides **189a**–**e**. The authors report this cascade as the synthetic equivalent of an acetaldehyde Mannich reaction. While the intermediate heterocycle was not isolated or explicitly discussed in this publication, it is implied that the 5-*exo* mode of cyclization predominates with the nascent thiovinyl radical. Collapse of the intermediate cyclized product following fluoride-mediated cleave provides the desired product in good yield. Friestad and Messari had also developed a diastereoselective vinyl addition to chiral hydrazones in an earlier account [84], and proceeded to develop a similar chiral hydrazone cyclization with silicon-tethered alkenes analogous to **187a**–**e** [85]. In the first of these accounts, vinyl addition to *α,β*-dihydroxyhydrazones was also investigated, with excellent extension of the scope of their methodology and without extensive protection strategies or hydroxyl group differentiation. The methodology employed by Friestad et al. hinges on the exploitation of temporary silicon-tethers, which permits facile and stereocontrolled vinyl addition to C=N bonds to furnish valuable synthetic scaffolds, such as acyclic amino alcohols [84].

Previously discussed work by Padwa et al. has shown that a sulfur atom suitably disposed to undergo S_H_i can extrude an acyl radical under the appropriate conditions. Almost twenty years after this account, Benati and co-workers exploited this methodology for the production of aldehydes under tin and silicon free conditions (Scheme 28) [86]. Benati’s radical cascade reaction between thiophenol and pentynylthiol esters **190** provides both aromatic and aliphatic aldehydes, and permits a formal reduction of esters to aldehydes even with substituents which would normally be very sensitive to any reducing conditions. The formation of concomitant dihydrothiophenes **191** frames this account as relevant to the present review, and offers a useful synthetic route to this sulfur-based heterocycle; albeit with a lack of diastereoselectivity.

In a more exotic example of radical annulation, a powerful method to access lactams was described by Tojino et al., which proceeded with incorporation of carbon monoxide **195** into the incipient heterocycle (Scheme 29) [87]. This cyclizative radical carbonylation of suitable azaenyne substrates **194** generated a number of nitrogenous heterocycles, with a comparison between silicon and thiol based radical precursors. *E/Z* selectivity was observed for the thiol and silane mediated cyclizations in which 100% *E*-selectivity was observed in many cases (e.g., **198a** and **198b**); this is in contrast to classical tin-based methods, which mostly initiate a *Z*-selective process [88].

Okiko Miyata and co-workers published an account in their series of radical cyclizations for heterocycle synthesis, in which selectivity for cyclization onto oxime ethers was reported (Scheme 30). This account details some of the previously discussed selectivity for radical addition to electron deficient alkenes, and the key advancement of this work is the generation of 5-membered lactams with alkyloxyamino moieties **200**, which is the result of preferential addition of an α-carbonyl radical to the oxime ether **199**. Analogous to the explanation in Section 2.2., the regioselectivity of this process can be explained by SOMO/HOMO interactions, and the stability of the radical intermediate [89].

A short communication from Nanni et al. on the regioselectivity of imidoyl radical cyclizations features aryl thiyl radicals, generated from photolytic cleavage of a disulfide (Scheme 31) [90]. This paper, published in the 2000s, expands upon original work conducted in the late 90s by the authors who could not unambiguously assign the identity of the products at the time [91]. This reaction between thiyl radicals and isocyanides comprises a cascade reaction in which radical **204** attacks the isocyanide group of **205**. Concomitant nitrogen-centered radical formation allows the cascade to continue through cyclization into the aromatic backbone **207**. The authors postulated that the fast, reversible cyclization of the imidoyl radical onto the sulfur center is offset by the slow but essentially irreversible radical addition to the cyano group. Two reaction pathways are proposed to explain the products observed, and the one to yield the major product **208** is shown below in Scheme 31.

In the same year, Rainer and Kennedy reported a cascade to dithioindoles **212a**–f from arylisonitriles bearing alkynes as *ortho* substituents **209 [92]**. A number of indoles were successfully synthesized, with the 5-*exo*-*dig* process appearing to outcompete other cyclization pathways (Scheme 32). A number of thiyl radical precursors were tested using a model substrate, and successful cyclization yields ranged from ca. 50 to 95%

A similar approach was published by Mitamura et al., relying on cyclization of the thioimidoyl radical onto an *ortho* alkene [93]. Radical cyclization of *ortho*-vinyl and *ortho*-allyl phenylisocyanides furnished bisthiolated indole and quinoline derivatives in moderate yields. This approach employed a photoinduced thiotelluration, and while tellurium is absent from the bisthiolated product, it appears to play a crucial role in the reaction mechanism; this is analogous to systems employing the diphenyl disulfide and diphenyl diselenide pair. It is probable that the tellurium/sulfur combination affords regioselectivity due to the high reactivity of the phenylthiyl radical [94], and the high carbon radical capturing ability of the phenyltelluride species. It is noteworthy to consider that the relative rate of dichalcogenides for capturing carbon-entered radicals is of the order *k*_PhSSPh_*:k*_PhSeSePh_*:k*_PhTeTePh_
*=* 1:160:630 [95,96,97].

Analogous to those results by Rainer and co-workers, Minozzi et al. published a similar radical cyclization of an alkynyl arylisonitrile **213** [98]. Interestingly, this proceeded as a multi-step cascade which resulted in the formation of quinoline **220** (Scheme 33, Insert A), as well as formation of spirocycles **218** and **219**. The authors propose that the thioimidoyl radical **214** formed from thiyl addition to the isonitrile can cyclize onto the alkyne as previously shown, but a 1,5-hydorgen migration and cyclization then furnishes the observed spirocyclic compounds. A proposed mechanism is detailed in Scheme 33.

### 3.4. Modern Approaches

Each new report that demonstrates the usefulness of thiyl radicals for cascade chemistry further broadens the array of substrates that can be successfully cyclized. In the following subsection, recent results from the last decade are discussed, with the intent of outlining the current state-of-the-art for heterogeneous cyclizations.

An excellent account of enyne cyclization by Taniguchi et al. builds upon work published forty years previous by Wang [99], in which the disulfide bond of diphenyl disulfide was cleaved by dimethylaniline to yield polymers via reaction with a suitable nitrile species. This original reaction was postulated to initiate by single-electron transfer (SET) from the amine. Taniguchi and co-workers then published a hydrothiolation of alkynes and subsequent cascade cyclization, with initiation of the process by tripropylamine (Scheme 34) [100]. This methodology offers an inexpensive approach to cyclization, and the annulated products can be separated by evaporation of the alkyl amine and standard chromatography. A mixture of 5-*exo* and 6-*endo* products were obtained, and the account provides an interesting example of aqueous thiyl radical addition, and non-classical initiation. A number of examples, with a range of olefinic substituents and varying heteroatoms bridging the alkyne and alkene were reported.

Access to the pentacyclic core of the fungal-derived asperparalines was explored by Crick et al., who designed a clever radical cascade sequence comprised of 1,6-hydrogen atom transfer, followed by two cyclization steps (Scheme 35) [101]. These valuable alkaloids have been shown to act as strong selective antagonists of nicotinic acetylcholine receptors [102] and possess a bridged bicyclic core which is the backbone of many natural products. While this account describes multiple cyclized products generated during the optimization of cyclization methodology, Crick and co-workers were able use diketopiperazine **230** to isolate a single diastereomer product **231** with the desired configuration for the asperparaline core, albeit in modest yield.

Shortly thereafter, Zhou and colleagues published a manganese (III) mediated addition of thiyl radicals to 1,3-diarylpropynones to yield thiolated indenones (Scheme 36). The account describes the use of Mn(OAc)_3_-induced formation of thiophenoxyl radical which undergo regioselective addition to an array of substrates, which can be further functionalized. The authors note that this offers an alternative to the heavy-metal catalysis (e.g., Pd or Rh complexes) employed to form 2,3-disubstituted indenones **234** from alkynes **232**, and permits mild conditions and moderate scope.

Previously we have seen the use of S_H_i chemistry to synthesize thiophenes from suitable diyne precursors. These are limited to substrates bearing homogenous unsaturation, and thus Kamimura et al. published a detail account [103] on the use of radical cascades to prepare bicyclic dihydrothiophenes (Scheme 37). Once again, the translocated radical **243** undergoes intermolecular addition at the sulfur atom, resulting in extrusion of an alkyl radical. A reaction pathway proceeding through 5-*exo*-*dig* cyclization and either stepwise or direct homolytic substitution was proposed. Overall, chiral enynes were converted to bicyclic pyrolidinodihydrothiophenes **244** in a process where a thiyl radical serves as a sulfur biradical surrogate. Improvements in stereoselectively were observed by Kamimura et al. when large excesses of the thiyl radical precursor were employed in a photoinitiated reaction. An assortment of alkyl disulfides were tested as thiyl radical precursors, and dimethyl disulfide was determined as the best precursor, with 78% yield and a 90:10 *trans*:*cis* ratio attained with 20 equivalents of disulfide. The authors also varied the substitution of the tertiary amine backbone, and furnished a number of novel compounds, with fair to good stereoselectivity.

Patel and co-workers published an extensive methodology for the synthesis of tricyclic and spirocyclic heterocycles based on a thiyl radical/isocyanide couple (Scheme 38) [104]. This alternative to tin-based methods provides access to pyrroloquinolines **246,** and spirocycles **247** and **248** from a common precursor, aiding the investigation of different heterocyclic motifs. Interestingly, the authors were able to select for 6-*endo* or 5-*exo* cyclization by altering the equivalents of radical initiator. Generally, one observes a trend of 5-*exo* favoured cyclization when concentration is low (ca 0.05 M) and the number of equivalents of AIBN is high (ca. 1.5 equivalents). In contrast, high concentration (0.1 to 0.2 M) and low loadings of initiator (ca. 0.3 equivalents) seem to favour 6-*endo* cyclization. As discussed by the authors, the kinetic 5-*exo* spirocycle predominates with increased quantities of initiator, indicating the nascent heterocycle is trapped and that the reaction is reversible. It follows that the thermodynamic 6-membered heterocycle predominates when the cyclization is slow, such as in the case where AIBN loading is reduced.

Many of the cyclization reactions of enynes discussed herein utilize thiophenol as the thiyl radical precursor; while this reagent has enjoyed a special place in sulfur radical addition/cyclizations, a number of diverse thiyl precursors have demonstrated utility in this area. One such example that boasts desirable biological properties such as lipophilicity and resistance to metabolic processes is the trifluoromethylthio group (SCF_3_). Qiu et al. recently published a trifluoromethylthiolation cyclization between 1,6-enynes and an aromatic substituent [105], which permits the construction of a number of polycyclic fluorene systems (Scheme 39). Improved yield and reaction control necessitated the addition of a terpyridine ligand **Ligand-1**, which was postulated to coordinate to the HMPA additive and reduce the redox potential of the silver trifluoromethylthiol; this likely reduces oxidative decomposition. A number of annulated products were synthesized, and representative examples of this methodology are shown in Scheme 39.

One account of particular interest was published by Chen and co-workers in early 2019, in which the authors used visible light (in the form of a green LED) and Eosin Y to synthesize 2-sulfenylindenones from the corresponding 1,3-diarylpropynones (Scheme 40) [106]. While this approach boasts environmental compatibility and cheap reagents, admittedly the reaction time is quite long (up to 36 h) and yields varied widely based on reaction solvent. However, a number of indenone species **259a**–**j** were prepared and a wide substrate scope was explored, showing the forbearance of the methodology to differing functional groups on both the aromatic backbone and phenylthiyl species. 

### 3.5. State of the Art

Thiyl radical mediated cyclization has been employed in the synthesis of a number of privileged scaffolds and complex substrates. One such example by Gharpure et al. [107], reports the synthesis of pyrrolo[1,2-*a*]indole derivatives. This motif appears in a number of important pharmaceutical compounds such as those possessing antimalarial properties like Flinderole C [108], and mitomycins which act as antitumor agents that can cross-link DNA [109]. This account by Gharpure and co-workers describes the use of enyne cyclization through a 5-*exo*-*trig* mechanism to generate a number of *N*-fused indolines **262a**–**c** and indoles **260a**–**c** from the corresponding *N*-propargylindones **261** (Scheme 41). The preference for indole vs. indolines was dictated by the identity of the propargylic substituents; indoline derivatives were furnished when R_2_ = Ph and R_3_ = H, and indoles were generated when these substituents were replaced by alkyl moieties such as a *gem*-dimethyl group. The authors propose that the bulkier aryl ring prefers to occupy the less sterically crowded face of the transition state. As the reduction of the radical (formed via 5-*exo*-trig cyclization onto the heterocyclic backbone) proceeds through a late transition state, the R_1_ substituent also prefers to occupy the less sterically incumbent face of the transition state, which permits delivery of a hydrogen atom to furnish the stereochemistry observed in **262a**–**c**. To explain the generation of **260a**–**c,** Gharpure et al. propose that the presence of an alkyl substituent on both faces of the transition state significantly slows down this hydrogen atom delivery, and thus when R_2_ and R_3_ = alkyl, a competing hydrogen abstraction provides the indole structures observed in **260a**–**c** with concomitant aromatization as the driving force.

The products obtained through this methodology were elaborated to the multicyclic core of yuremamine, a natural product featuring five contiguous stereocenters [110]. Shortly thereafter, Shibata and co-workers described the thiyl radical mediated cyclization of ω-alkynyl *O*-silyloximes with an odourless thiyl precursor, 4-*tert*-butylbenzenethiol [111]. This approach furnished a number of cyclic *O*-silylhydroxylamines in good yields. Removal of the silyl group yielded a number of hydroxylamines which could be further transformed to nitrones, a motif which has demonstrated potential in neuroprotective and antiaging drugs [112].

While many of the examples discussed herein have employed thermal initiation in boiling benzene, or photolysis with high intensity ultraviolet lamps, there are a number of visible-light induced cascades reported in the literature. Xie et al. developed a reliable route to benzothiophenes (Scheme 42A) through the visible-light induced cyclization of 2-alkynylanilines **263** at room temperature [113]. Good yields and overall forbearance to substitution on the alkyne terminus are reported, and a mechanism in which hydrogen peroxide oxidizes cyclized intermediate **270** to reform aromaticity is proposed. Analogous to the control experiments by their contemporaries, the authors employed a ‘TEMPO test’, in which one equivalent of TEMPO is added to the reaction, and the quenched radical intermediates identified by mass spectrometry. A similar approach was published by Ye et al. [114], which again relied on visible light to cyclize alkynes **273** and aryldisulfides **266** to benzothiophenes **274** (Scheme 42B). Interestingly, the authors found that oxygen could be used as the sole oxidant, and irradiation with sunlight could initiate the reaction efficiently. Ye and co-workers identified the optimal concentration of O_2_ to be around 15% and explained that although the reaction also occurs under argon, this is likely due to oxygen seeping through the septum. Yuan et al. achieved similar findings (Scheme 42C) to Ye et al. and Xie et al., while exploring the use of *N*-methacryloylbenzamides **275** as substrates to generate isoquinoline-type compounds **277** [115]. The contributions from the three aforementioned authors are summarized in below, and the putative mechanism proposed by Xie et al. is included.

Shortly thereafter, Nair et al. described a diastereoselective visible-light mediated cascade of cyclohexadienones **278** with alkyne tethers (Scheme 43) [116]. Good yields were reported, as was good functional group tolerance. This account relies on Eosin Y as the organic photocatalyst, and boasts an array of thiophenols/thiols as radical precursors, with electron donating and electron withdrawing groups in *para*- and *ortho*- positions tolerated for the aromatic thiols. The authors were also able to demonstrate the applicability to gram scale synthesis, and reported favourable green chemistry metrics (atom economy, and efficiency). A publication [117] by Mallick et al. reported comparable results, employing the same reactions conditions previously developed in the Sahoo group (Section 2.4, Dutta et al.). Mallick and co-worker’s approach employed a *N*-hydroxyphthalimide-mediated radical cyclization of yne-dienones, yielding 3-thioaryl-[6,6]-dihydrochromenone derivatives. Altering the substituents of the aryl thiol, or the functional groups presented by the yne-dienone were again well tolerated in the cascade.

Further examples of the use of isocyano groups as radical acceptors was published by Li et al. in 2019, in which a denitrogenative annulation of 1-azido-2-isocyanarenes **289** furnished an array of 2-thiolated benzimidazoles **291** (Scheme 44) [118]. The authors report both a ‘promoter-free’ pathway and one involving AIBN, both of which are heated in toluene in an inert atmosphere. While the AIBN pathway offers higher yields in almost every example, a catalyst-free pathway in which the only by-product is the extruded nitrogen gas from cyclization onto the *ortho*-azide is appealing. Li et al. argue that that the thiyl radicals could be initiated by trace oxygen in the reaction vessel, although it has previously been reported that certain thiols can react successfully via a radical mechanism without catalysts or initiators [119], and indeed thiols may react with isocyanides at elevated temperatures [120]. The authors report this one pot method as having high functional group tolerance, and a number of prepared examples are shown in Scheme 44.

Research from Gharpure et al. elaborates on the pyrrole synthesis previously developed by Dutta et al., but instead offers tetrasubstituted furans **297a**–**c** from alkynyl vinylogous carbonates **296** (Scheme 45) [121]. The authors note that substitution at the α-position of the alkynyl vinylogous carbonate affects diastereoselectivity of the cascade, and propose a mechanism in which the thiyl radical attacks the alkyne first. The nascent carbon-centered radical then cyclizes onto the remaining olefin, and the cascade continues with propagation of the radical chain. It is noteworthy that in this account, thiophenol can act not only as a source of thiyl radicals but as an oxidant with AIBN to transform dihydrofurans to furans.

The Hong group reported a breakthrough in mid-2019, in which the substrate and photosensitizer were combined in a single molecule (Scheme 46). This account by Kim et al. examined the photochemical activity of quinolinone-based substrates **304** which could reach an excited state and initiate thiyl radical formation, while the ground state acted as a radical accepting substrate. The products of this ground state substrate are dictated by whether a hydrogen atom transfer (HAT) or single electron transfer (SET) step occurs, which is determined by the coupling reagents used [122]. The authors report a diverse assemblage of dihydro- and tetrahydrophenanthridin-6(5H)-ones prepared in a metal-free and essentially photocatalyst-free cascade. A number of bioactive thiols such as cysteine, captopril and gemfibrozil were successfully applied in this methodology, which exerts a 6-*endo*-*trig* cyclization following thiyl radical addition. The tolerance of this methodology to bioactive thiols indicates that this method may be applied to generating compound libraries for combinatorial chemistry (or otherwise), or modifying existing bioactive thiols. A selection of examples from both the disulfide and thiol pathways are shown in Scheme 46.

Somewhat analogous to the work reported by the Hong group, a recent report by Ho et al. describes an electron-donor-acceptor (EDA) complex which undergoes visible light-induced charge transfer, and facilitates the radical spirocyclization of a number of indole-tethered ynones **307a**–**j** (Scheme 47) [123]. The EDA complex promotes thiyl radical generation and initiates a cascade that features a dearomatizing spirocyclization with concomitant carbon-sulfur bond formation, to yield a number of indolines **308a**–**j**. The authors highlight that this is only the second reported use of intramolecular EDA complexes in the literature [124], and admit that this rare form of radical activation was uncovered serendipitously. Regioselective addition of the somewhat electrophilic thiyl radicals to ynones initiates the process, followed by attack of incipient carbon-centered radical on the tethered aromatic group.

## 4. Conclusions

Thiyl and acyl thiyl radicals have found broad application in organic synthesis, from heterocycle formation to sterically recalcitrant carbocyclizations. They boast mild generation and remarkable reactivity; their utility provides access to radical chemistry to those chemists beyond the archetypal sulfur laboratory. The addition of these radicals to homogenous systems facilitates the synthesis of alicycles and thiophenes, while their addition to heterogeneous systems gives rise to numerous heterocycles and fused ring systems. Their application to intramolecular cascade reactions has been widely explored, and continues to be developed for organocatalytic and enantioselective systems. This review highlights the vast array of complex architectures rendered accessible by the thiol-ene and thiol-yne reactions in particular, and hopes to demonstrate the versatility of thiyl and acyl thiyl radicals as a whole.

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
