# Peer review of "Thiyl Radicals: Versatile Reactive Intermediates for Cyclization of Unsaturated Substrates"

_molecules, 2020, doi:10.3390/molecules25133094_

Round 1

Reviewer 1 Report

The present review focuses on selected recent examples of cyclization reactions using thiyl radicals. Although there have already been comprehensive reviews on thiyl radicals, I agree that this manuscript will be beneficial to readers of this journal because the authors made their effort to avoid excessive overlapping with previous reviews. Therefore, I recommend publication of this manuscript as a review after minor revisions.

1) Please show simple mechanisms for some complex transformations. For example, Schemes 6, 12, 13, 28, 29, 46.

2) Scheme 14, please show the structure of the catalyst.

3) Scheme 18, would 6-membered product 115 be given by direct 6-endo cyclization or 5-exo followed by rearrangement?

4) Scheme 41, please show a reason why different products were obtained depending on reaction conditions.

Author Response

1) Please show simple mechanisms for some complex transformations. For example, Schemes 6, 12, 13, 28, 29, 46.

Mechanisms have been included in each of these schemes, showing the expected radical intermediates and pathways to the expected product.

Regarding Scheme 46, lines 783-785 have been updated to: “This account by Kim et al. examined the photochemical activity of quinolinone-based substrates 304 which could reach an excited state and initiate thiyl radical formation, while the ground state acted as a radical accepting substrate. The products of this ground state substrate are dictated by whether a hydrogen atom transfer (HAT) or single electron transfer (SET) step occurs, which is determined by the coupling reagents used.[122]” in order to put the mechanism in context.

2) Scheme 14, please show the structure of the catalyst.

The structure of the catalyst optimised by Hashimoto et al. has been included.

3) Scheme 18, would 6-membered product 115 be given by direct 6-endo cyclization or 5-exo followed by rearrangement?

The authors of the original work considered this, and had consulted studies by both Stork and Beckwith. Broka et al. proposed an explanation in their 1987 account, and thus we have included the following text to lines 356-358 of the review: “Broka et al. proposed that equilibration of the cyclopentylmethyl radical, formed via 5-exo-trig cyclization of 114, to the more stable cyclohexyl isomer prior to hydrogen abstraction facilitates the generation of 115.”

4) Scheme 41, please show a reason why different products were obtained depending on reaction conditions.

The identity of the propargylic substituents dictates the product identity. Scheme 41 has been updated to show “If R2 = Ph and R3 = H” under one reaction arrow, and “If R2 and R3 = alkyl” under the other reaction arrow. 

The following explanation has also been included at lines 693-704, to further elaborate upon the schematic update: “The preference for indole vs. indolines was dictated by the identity of the propargylic substituents; indoline derivatives were furnished when R2 = Ph and R3 = H, and indoles were generated when these substituents were replaced by alkyl moieties such as a gem-dimethyl group. The authors propose that the bulkier aryl ring prefers to occupy the less sterically crowded face of the transition state. As the reduction of the radical (formed via 5-exo-trig cyclization onto the heterocyclic backbone) proceeds through a late transition state, the R1 substituent also prefers to occupy the less sterically incumbent face of the transition state, which permits delivery of a hydrogen atom to furnish the stereochemistry observed in 262a-c. To explain the generation of 260a-c, Gharpure et al. propose that the presence of an alkyl substituent on both faces of the transition state significantly slows down this hydrogen atom delivery, and thus when R2 and R3 = alkyl, a competing hydrogen abstraction provides the indole structures observed in 260a-cwith concomitant aromatization as the driving force.”

Reviewer 2 Report

The authors presented a very nice review of thiyl radical mediated cyclization reactions. The manuscript was well organized and focused on two types of reactions, i.e., addition to homogeneous and heterogeneous substrates. The account is in a chronicle order which is very easy to follow and very helpful for readers to understand the progress in this field. No significant flaws were found in this manuscript. It is recommended to publish as it is. 

Author Response

We thank the referee for their positive feedback, no corrections were required.

Reviewer 3 Report

This Review article by Lynch and Scanlan focusses on the use of thiyls radicals in the cyclization of unsaturated substrates. I have really enjoyed reading this contribution which is very useful for the field. I recommend acceptance in Molecules.

Author Response

(The authors gave the same response as above.)

Reviewer 4 Report

The text associated with Scheme 27 needs a reference citation.

Friestad, G. K.; Jiang, T.; Fioroni, G. M.  "Tandem Thiyl Radical Addition and Cyclization of Chiral Hydrazones Using a Silicon-Tethered Alkyne." Tetrahedron: Asymmetry 2003, 14, 2853-2856.

The same section should also be expanded to include the following:

Friestad, G. K.; Massari, S. E.  "Diastereoselective Vinyl Addition to Chiral Hydrazones via Tandem Thiyl Radical Addition and Silicon-Tethered Cyclization." Org. Lett. 2000, 2, 4237-4240.

Friestad, G. K.; Massari, S. E.  "A Silicon Tether Approach for Addition of Functionalized Radicals to Chiral alpha-Hydroxyhydrazones: Diastereoselective Additions of Hydroxymethyl and Vinyl Synthons." J. Org. Chem. 2004, 69, 863-875.

Author Response

The text associated with Scheme 27 needs a reference citation.

Friestad, G. K.; Jiang, T.; Fioroni, G. M.  "Tandem Thiyl Radical Addition and Cyclization of Chiral Hydrazones Using a Silicon-Tethered Alkyne." Tetrahedron: Asymmetry 2003, 14, 2853-2856.

This citation has been added.

The same section should also be expanded to include the following:

Friestad, G. K.; Massari, S. E.  "Diastereoselective Vinyl Addition to Chiral Hydrazones via Tandem Thiyl Radical Addition and Silicon-Tethered Cyclization." Org. Lett. 2000, 2, 4237-4240.

Friestad, G. K.; Massari, S. E.  "A Silicon Tether Approach for Addition of Functionalized Radicals to Chiral alpha-Hydroxyhydrazones: Diastereoselective Additions of Hydroxymethyl and Vinyl Synthons." J. Org. Chem. 2004, 69, 863-875.

Both of these citations have now been added, and the work therein briefly discussed between lines 498-505. The addition reads as follows: “Friestad and Messari had also developed a diastereoselective vinyl addition to chiral hydrazones in an earlier account,[84] and proceeded to develop a similar chiral hydrazone cyclization with silicon tethered alkenes analogous to 187a-e.[85] In the first of these accounts, vinyl addition to α,β-dihydroxyhydrazones was also investigated, with excellent extension of the scope of their methodology and without extensive protection strategies or hydroxyl group differentiation. The methodology employed by Friestad et al. hinges on the exploitation of temporary silicon-tethers, which permits facile and stereocontrolled vinyl addition to C=N bonds to furnish valuable synthetic scaffolds, such as acyclic amino alcohols.[84]